# Unaffected fellow eye neovascularization in patients with type 3 neovascularization: Incidence and risk factors

Jae Hyuck Kwak[1,2], Woo Kyung Park[1,2], Rae Young Kim[1,2], Mirinae Kim[1,2], Young-Gun Park[1,2], Young-Hoon Park[1,2]*

1 Department of Ophthalmology and Visual Science, Seoul St. Mary's Hospital, College of Medicine, The Catholic University of Korea, Seoul, Korea, 2 Catholic Institute for Visual Science, College of Medicine, The Catholic University of Korea, Seoul, Korea

* parkyh@catholic.ac.kr

**Data Availability Statement:** All relevant data are within the manuscript and its Supporting information files.

## Abstract

### Purpose

To evaluate the incidence and risk factors of neovascularization in unaffected fellow eyes of patients diagnosed with type 3 neovascularization in Korea.

### Methods

This retrospective study included 93 unaffected fellow eyes of 93 patients diagnosed with type 3 neovascularization. For initial type 3 neovascularization diagnosis, optical coherence tomography and angiography were conducted. These baseline data were compared between patients with and without neovascularization in their fellow eyes during the follow-up period.

### Results

The mean follow-up period was 66.1±31.1 months. Neovascularization developed in 49 (52.8%) fellow eyes after a mean period of 29.5±19.6 months. In the fellow eye neovascularization group, the incidence of soft drusen and reticular pseudodrusen was significantly higher than that in the non-neovascularization group (83.7% vs. 36.5%, p<0.001; 67.3% vs. 40.9%, p = 0.017, respectively), but the choroidal vascularity index (CVI) showed a significantly lower value (60.7±2.0% vs. 61.7±2.5%; p = 0.047). The presence of reticular pseudodrusen was related with the duration from baseline to development of fellow eye neovascularization (p = 0.038).

### Conclusion

Neovascularization developed in 52.8% of unaffected fellow eyes. The presence of soft drusen, reticular pseudodrusen, and lower CVI values can be considered risk factors of neovascularization in unaffected fellow eyes of patients with type 3 neovascularization. The lower CVI values suggest that choroidal ischemic change may affect the development of choroidal neovascularization in these patients.

**Funding:** This study was supported by the Basic Science Research Program through the National Research Foundation of Korea (NRF-2020R1F1A1074898). URL of funder website: https://www.nrf.re.kr. Initials of the authors who received each award: YHP. The funders had no role in study design, data collection and analysis, decision to publish, or preparation of the manuscript.

**Competing interests:** The authors have declared that no competing interests exist.

## Introduction

Retinal angiomatous proliferation (RAP) is a type of neovascular age-related macular degeneration (nAMD). Yannuzzi et al [1] first proposed the term RAP in 2001 and described that in this disease, the neovascular membranes originate from the proliferation of intraretinal capillaries, that is related with a telangiectatic response. The main distinction between RAP and other types of nAMD is that RAP is characterized by the intraretinal location of the neovessels. Therefore, Freund et al proposed an alternative term "type 3 neovascularization" for this type of nAMD to emphasize the location of the neovascular complex independent of its origin [2]. The frequency of type 3 neovascularization has been reported to be 15–20% among nAMD cases in Caucasians [2–4], and 4.5–11.1% in Asian patients [5–7].

Numerous studies have reported that type 3 neovascularization has a poor prognosis and requires appropriate treatment to prevent disease progression [8–10]. In addition, many studies have suggested that type 3 neovascularization tends to bilateral involvement [11–13]. Gross et al [12] reported that fellow eye type 3 neovascularization developed within 3 years in 100% of patients with unilateral type 3 neovascularization. In a recent study, Chang et al [13] reported an incidence of 38.3% of unaffected fellow eye neovascularization in patients with type 3 neovascularization during mean 27.8-month follow-up. These inconsistencies in the reported results could be due to variation in reporting baseline characteristics and follow-up period among studies. Nevertheless, all these studies indicate a high risk of unaffected fellow eye involvement in patients with type 3 neovascularization. Therefore, identifying the risk factors for fellow eye neovascularization is crucial for preventing bilateral poor visual outcomes in these patients.

Although epidemiologic studies of the risk factors for type 3 neovascularization are lacking, recent studies have shown that reticular pseudodrusen [13,14], larger drusen density, and thinner subfoveal choroidal thickness [15–17] were related with type 3 neovascularization.

Controversies remain regarding the origin and pathophysiology of type 3 neovascularization. Yannuzzi et al [4] suggested that type 3 neovascularization may consist of two neovascular change sites, one is in the retina and the other located in the choroid. Compared to the choroid, the retina is relatively easy to analyze through various reliable image tools, so several in vivo studies have confirmed the intraretinal origin of the initial angiogenic process [18–20]. However, unlike the retina, evaluating the choroid is difficult due to the lack of reliable analytic tools. Recently, the subfoveal choroidal thickness (SFCT) [21–23] and choroidal vascularity index (CVI) [24] have been employed as useful optic coherence tomography (OCT)-based parameters to quantify structural and vascular alterations of the choroid.

The purpose of this study was to evaluate the incidence of unaffected fellow eye neovascular change in Korean type 3 neovascularization patients and identify various risk factors, including choroidal quantitative parameters that may affect the development of fellow eye neovascularization.

## Methods

This study followed the tenets of the Declaration of Helsinki and was approved by the Institutional Review Board of Seoul St. Mary's Hospital, The Catholic University of Korea. Informed consent was waived due to the retrospective nature of this study.

We retrospectively reviewed the medical records of all treatment-naive Korean patients who were diagnosed with unilateral type 3 neovascularization at the Department of Ophthalmology and Visual Science of Seoul St. Mary's Hospital between January 2008 and March 2018 and were followed up for more than 2 years. On the first visit, complete ophthalmic examination was performed, including measurement of best-corrected visual acuity, slit-lamp

biomicroscopy, fundus photography, spectral-domain OCT (SD-OCT), fluorescein angiography (FA), and indocyanine green angiography (ICGA) (Spectralis HRA+OCT; Heidelberg Engineering, Heidelberg, Germany). All patients in this study were diagnosed and determined stage by two retinal specialists (W.K.P. and Y.G.P.) based on the findings of funduscopy, SD-OCT, FA, and ICGA. Yannuzzi's classification method [25] was applied in this study to determine stage. Not all fellow eyes showed any other form of neovascular maculopathy or macular edema at the initial examination. Patients with high myopia and other eye diseases, such as advanced degenerative macular disorders, epiretinal membrane, macular hole, retinal vascular obstruction, central serous chorioretinopathy, and uveitis in the fellow eye were excluded from our study.

The presence of drusen was determined using fundus photos and SD-OCT. Large soft drusen (>125 μm) and reticular pseudodrusen were confirmed by SD-OCT. The SFCT was measured as the vertical distance from the hyper reflective line corresponding to Bruch's membrane beneath the retinal pigment epithelium under the fovea to the inner scleral border using the software-based calipers of the OCT viewer. To measure the CVI, we adapted the OCT B-scan image based technique proposed by Agrawal et al. [26] In brief, image binarization and processing were performed using Image J software (version 1.52; provided in the public domain by the National Institutes of Health, Bethesda, MD, USA; http://imagej.nih.gov/ij/). In this technique, Niblack autolocal threshold was applied, and other compensation modes were not used. All images and measurements were evaluated by two retinal specialists (J.H.K. and Y.H.P.), who were masked to fellow eye information and angiography. In case of disagreement, a third retinal specialist (M.K.) finally determined the results.

The patients were followed up every 1 to 3 months according to their therapeutic plan. Best-corrected visual acuity measurement, fundus photography, and SD-OCT were conducted as a routine process. When neovascularization was suspected due to newly detected subretinal fluid, macular edema, retinal hemorrhage, and pigment epithelial detachment, FA and ICGA were performed to confirm the presence of neovascularization.

We divided the patients into two groups based on the development of neovascularization in the fellow eye during the follow-up period. The patients' baseline characteristics, including age, sex, follow-up duration, history of diabetes mellitus or hypertension, presence of large soft drusen and reticular pseudodrusen in the fellow eyes, and the SFCT and CVI in the fellow eyes at the initial ophthalmic examination, were compared between the two groups.

For Statistical analysis, we use IBM SPSS software (IBM Corp., Armonk, NY, USA). The independent t-test, chi-square test, and Fisher's exact test were employed to compare the baseline data between the two groups. The duration from baseline to development of fellow eye neovascularization, with or without risk factors, was compared using independent t-test. The cumulative proportions of fellow eye neovascularization-free patients, with and without large soft drusen and reticular pseudodrusen, was compared using Kaplan-Meier analysis and Log rank test. Continuous data are expressed as mean ± standard deviation. The results were considered statistically significant at p-value <0.05.

## Results

A total of 93 patients (18 men and 75 women) with a mean age of 77.2±6.0 years were included in this study. The mean follow-up duration was 66.1±31.1 months. Large soft drusen and reticular pseudodrusen were detected in more than 50% of fellow eyes (61.3% and 54.8%, respectively). The mean SFCT was 162.4±66.7 μm and the mean CVI was 61.2± 2.3%. The patients' baseline characteristics and clinical features are presented in Table 1.

**Table 1. Baseline characteristics and clinical features of patients presenting with unilateral type 3 neovascularization.**

| Baseline characteristics and clinical features | n = 93 |
|---|---|
| Age, years | 77.2±6.0 |
| Sex, male (%) | 18 (19.4%) |
| Follow-up duration, months | 66.1±31.1 |
| Diabetes mellitus, no.(%) | 16 (17.2%) |
| Hypertension, no.(%) | 32 (34.4%) |
| Large soft drusen of the fellow eye, no.(%) | 57 (61.3%) |
| Reticular pseudodrusen of the fellow eye, no. (%) | 51 (54.8%) |
| Subfoveal choroidal thickness of the fellow eye, μm | 162.4±66.7 |
| Choroidal vascularity index of the fellow eye, % | 61.2±2.3 |
| Type 3 neovascularization stage, no. (%) | |
| 1 | 21(22.6%) |
| 2 | 33(35.5%) |
| 3 | 39(41.9%) |

All continuous data are presented as means ± standard deviations.

Forty-nine patients (52.8%) developed neovascularization in the unaffected fellow eye during the follow-up period. The mean duration from baseline to development of fellow eye neovascularization was 29.6±19.6 (range 5–95) months. The cumulative incidence of neovascularization in the fellow eyes was 10.8% at 1 year and 23.7% at 2 years. Among the 49 fellow eyes that developed neovascularization, 42 (85.7%) presented with type 3 neovascularization and 7 had other types of nAMD. A representative case of type 3 neovascularization development in the fellow eye is shown in Fig 1.

The comparison data of baseline characteristics and clinical features between patients, with and those without fellow eye neovascularization to find the risk factors related with development of neovascularization in fellow eyes, is summarized in Table 2. The incidence of soft drusen and reticular pseudodrusen was significantly higher in the neovascularization group than in the non-neovascularization group ($p<0.001$ and $p = 0.017$, respectively). However, the CVI showed a significantly lower value in the neovascularization group ($p = 0.047$). Other risk factors, including age ($p = 0.606$), diabetes mellitus ($p = 0.091$), hypertension ($p = 0.350$), SFCT ($p = 0.176$), and baseline type 3 neovascularization stage ($p = 0.980$), showed no significant differences between the two groups.

Among the factors that showed significant differences between the groups, only reticular pseudodrusen was associated with the duration from baseline to development of fellow eye neovascularization ($p = 0.038$). In patients with reticular pseudodrusen in the neovascularization group, the mean duration was 25.6±15.5 months.

The cumulative proportions of fellow eye neovascularization-free patients, with and without large soft drusen, and reticular pseudodrusen are shown in Fig 2. The mean neovascularization-free period was 53.2±8.0 and 97.2±7.1 months in fellow eyes, with and without large soft drusen, respectively ($p<0.001$). In addition, this period was 52.7±6.6 months in the presence of reticular pseudodrusen and 100.1±13.2 months in the absence of reticular pseudodrusen ($p = 0.004$).

## Discussion

In this study, we investigated the incidence and risk factors of neovascularization in unaffected fellow eyes in Korean unilateral type 3 neovascularization patients. During follow-up period,

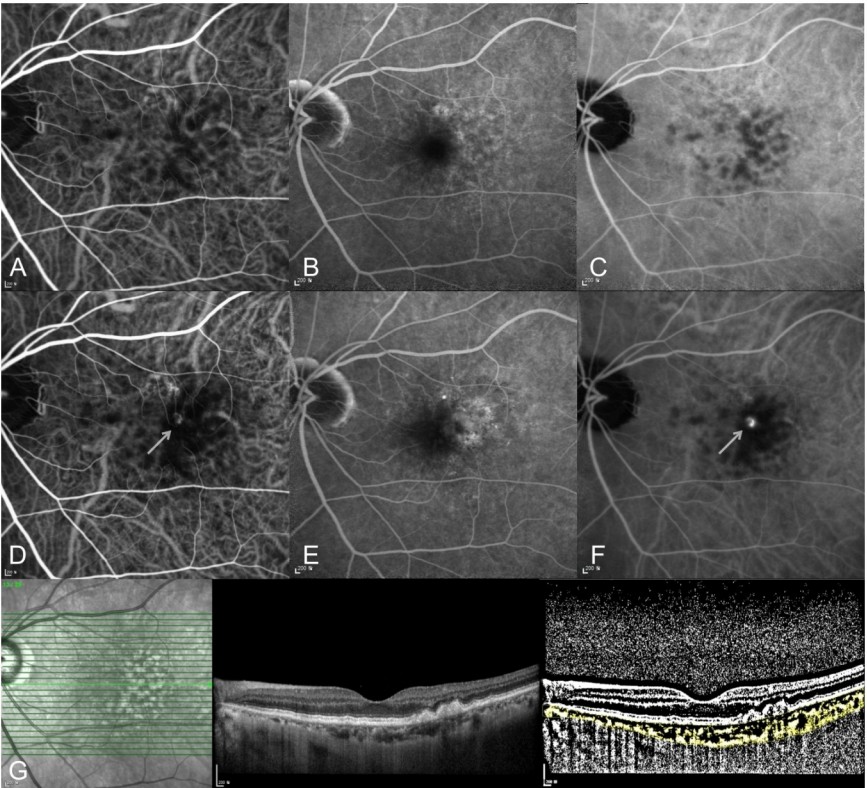

**Fig 1. A 72-year-old woman with right eye type 3 neovascularization.** (A-C) At initial diagnosis, no neovascular lesion was detected in her left eye at early-to-late phase on indocyanine green and fluorescein angiography. (A) early phase indocyanine green angiography, (B) late phase and fluorescein angiography, (C) late phase indocyanine green angiography. (D-F) Ten months later, type 3 neovascularization progressed in her left eye. The arrow indicates the neovascular lesion on indocyanine green angiography. (D) early phase indocyanine green angiography, (E) late phase and fluorescein angiography, (F) late phase indocyanine green angiography. (G) OCT B-scan and binarized image to measure the choroidal vascularity index (CVI) in the left eye at initial diagnosis of type 3 neovascularization in the right eye.

49 of 93 patients (52.8%) developed neovascular changes in the unaffected fellow eyes, and the mean interval to binocular involvement was 29.6 months. The presence of large soft drusen and reticular pseudodrusen, as well as lower CVI values, were significantly correlated with the incidence of fellow eye neovascularization.

Neovascularization developed in more than half of unaffected fellow eyes in our study, and this result shows that type 3 neovascularization tends to involve bilaterally. This presents a similar trend as the results reported in previous studies [11–13], but there is a difference in the rate of fellow eye involvement. Since few similar studies have been conducted in Asians, the answer to the inconsistency in incidence of unaffected fellow eye neovascularization is thought to be found in subsequent studies.

Regarding the risk factors for fellow eye neovascularization in type 3 neovascularization, previous studies have reported that a larger drusen density, thinner subfoveal choroidal thickness [15–17], and reticular pseudodrusen [13,14] were associated with the incidence of RAP. More recently, Masaaki et al. [27] reported that hypoautofluorescence on near-infrared autofluorescence imaging may be associated with early-stage type 3 neovascularization. Furthermore, a large soft drusen (≥125 μm in diameter) is a well-known risk factor for nAMD [28]. Our results tend to support the results of these previous studies. The mean time to bilateral

**Table 2. Factors associated with neovascularization in the fellow eye of patients presenting with unilateral type 3 neovascularization.**

| Risk factors | Fellow eye NV(+) n = 49 | Fellow eye NV(-) n = 44 | p-value |
|---|---|---|---|
| Age, years | 76.9 ± 5.5 | 77.6 ± 6.5 | 0.606[†] |
| Sex, male (%) | 9 (18.4%) | 9 (20.5%) | 0.799[*] |
| Follow-up duration, months | 68.4 ± 31.4 | 63.5 ± 31.0 | 0.458[†] |
| Diabetes mellitus, no.(%) | 12 (24.5%) | 4 (9.1%) | 0.091[‡] |
| Hypertension, no.(%) | 19 (38.8%) | 13 (30.0%) | 0.350[*] |
| Large soft drusen of the fellow eye, no. (%) | 41 (83.7%) | 16 (36.46%) | <0.001[*] |
| Reticular pseudodrusen of the fellow eye, no. (%) | 33 (67.3%) | 18 (40.9%) | 0.017[*] |
| Subfoveal choroidal thickness of the fellow eye, μm | 156.9 ± 53.4 | 167.9 ± 78.3 | 0.176[†] |
| Choroidal vascular index of the fellow eye, % | 60.7 ± 2.0 | 61.7 ± 2.5 | 0.047[†] |
| Type 3 neovascularization stage, no. (%) | | | 0.980[*] |
| 1 | 11 (22.4%) | 10 (22.7%) | |
| 2 | 17 (34.7%) | 16 (36.4%) | |
| 3 | 21 (42.9%) | 18 (40.9%) | |

All continuous data are presented as means ± standard deviations.

[†] p-value by independent t-test,

[*] p-value by chi-square test,

[‡] p-value by Fisher's exact test,

NV, neovascularization.

involvement in patients with these types of drusen was also consistent with the trends reported in previous studies. However, to the best of our knowledge, few studies have analyzed the CVI in patients with type 3 neovascularization. Moreover, type 3 neovascularization predominantly involves both eyes; thus, the CVI analysis in the unaffected fellow eye conducted in this study would be valuable.

Since some histopathological studies reported that the choroid may play a significant role in the pathogenesis of nAMD [29–32], investigation of structural and vascular alterations in the choroid has been a major issue to understand nAMD. However, due to the lack of reliable imaging tools capable of sufficient evaluation in clinical conditions, OCT-based parameters, including SFCT and CVI, have been considered as a valuable alternative due to their noninvasive nature and ease of quantification. According to Agrawal et al. [33] the CVI was reduced in the affected eye of patients with nAMD despite the unchanged choroidal thickness because the CVI may be more sensitive in the analysis of choroidal vascular components. In the current

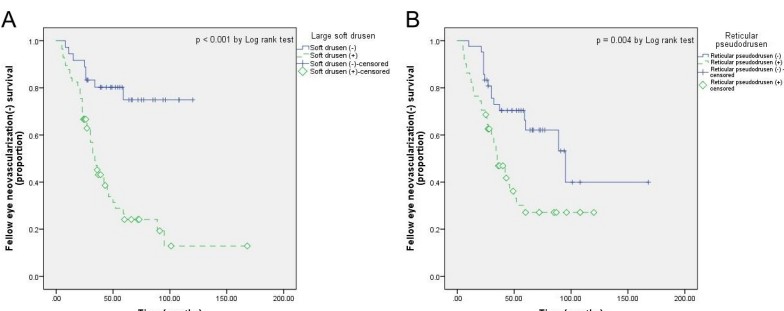

**Fig 2. Kaplan-Meier curve analysis of fellow eye neovascularization-free patients.** Comparison of Kaplan-Meier curves indicating the cumulative proportion of fellow eye neovascularization-free patients among those with and these without (A) large soft drusen and (B) reticular pseudodrusen.

study, we found that a lower CVI may be associated with the development of neovascularization in the unaffected fellow eye of type 3 neovascularization patients. Reduced choroidal vascularity could lead to choroidal ischemic change, and this condition might affect the formation of choroidal neovascularization. According to a recently published study, melanin deficiency may occurs in retinal pigment epithelium cells as a result of intensive macular stress, and there is a possibility that type 3 neovascularization may occur due to several cytokines including vascular endothelial growth factors secreted during this period [27]. Here, intensive macular stress can be induced from ischemic choroidal condition, so our study is thought to show a trend that is not different from the results of previous studies. Thus, these results show the possibility that low CVI values in the unaffected fellow eyes of patients with type 3 neovascularization may be associated with progressed choroidal ischemic change and possible subclinical disease. Based on the results of the current study, we expect that the early evaluation of the CVI in the unaffected fellow eyes of patients with type 3 neovascularization may help in establishing a proper follow-up plan.

The mean SFCT of the fellow eye in unilateral RAP was 162.4 μm in this study, and the difference between the two groups was not statistically significant. Most of the nAMD types that developed in the fellow eye were type 3 in our study (85.7%), but the mean SFCT of the fellow eyes was not significantly lower in the group with neovascularization. The reason for this result is thought to be that the SFCT can be influenced by multiple factors, including axial length, age, and intraocular pressure [34]. Therefore, it would be difficult to evaluate the SFCT as a single risk factor for predicting development of fellow eye neovascular change in unilateral type 3 neovascularization.

To the best of our knowledge, this is the first study investigating the influence of the CVI in the development of fellow eye neovascularization in patients with unilateral type 3 neovascularization. However, our study had several limitations. First, this study was conducted as a single center study. A single center study with limited number of study collective may have limited power to verify incidence of the neovascularization of the fellow eyes related type 3 neovascularization in Korean population. Second, the measurement of the SFCT and CVI was performed manually because there is no automated software. In addition, in the current CVI theory, dark and white pixels in the binarized image were assumed to present the vascular and stromal structures, which were not histologically confirmed. However, there is no definite evidence to rebut this theory and previous empirical studies strongly support this theory [33]. Third, optical coherence tomography angiography (OCTA) was not included as routine ophthalmic examination in this study. In cases without retinal fluid, early type 3 neovascularization lesion may be present on OCTA. Further studies including OCTA findings are needed and will provide more information to determine risk factors. Finally, as a result of this study, the mean difference between the CVI values in the two groups was 1%, which was not large, and the p-value also showed that the significance level was not very high (p = 0.047). However, based on the results of this study, we believe that the results of this study can be reinforced by conducting further studies with well-calculated sample sizes.

In conclusion, the incidence of neovascularization in the unaffected fellow eyes of patients with unilateral type 3 neovascularization was 52.8% during the follow-up period ($\geq$ 2 years) and the mean interval to bilateral involvement was 29.6 months. Determined factors related with the development of neovascularization in the fellow eyes were large soft drusen and reticular pseudodrusen. Moreover, this study demonstrated that choroidal ischemic change of the fellow eye, confirmed by a lower CVI value, might affect choroidal neovascularization. At present, the CVI is measured manually, but if an automated CVI analysis tool becomes available, we expect that lower CVI values in the fellow eye could be considered as a useful indicator of risk for neovascularization, which can be easily referenced in clinical practice.

## Supporting information

**S1 File.**
(XLSX)

## Author Contributions

**Conceptualization:** Jae Hyuck Kwak, Young-Hoon Park.

**Data curation:** Woo Kyung Park, Rae Young Kim, Mirinae Kim, Young-Gun Park.

**Formal analysis:** Jae Hyuck Kwak, Woo Kyung Park, Rae Young Kim, Mirinae Kim, Young-Gun Park.

**Funding acquisition:** Young-Hoon Park.

**Investigation:** Young-Hoon Park.

**Methodology:** Young-Hoon Park.

**Project administration:** Jae Hyuck Kwak, Young-Hoon Park.

**Supervision:** Young-Gun Park, Young-Hoon Park.

**Writing – original draft:** Jae Hyuck Kwak.

**Writing – review & editing:** Young-Hoon Park.

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
