## [Decision Letter · Decision Letter 0]

30 Mar 2021

PONE-D-21-05303

Unaffected fellow eye neovascularization in patients with retinal angiomatous proliferation: incidence and risk factors

PLOS ONE

Dear Dr. Park,

Thank you for submitting your manuscript to PLOS ONE. After careful consideration, we feel that it has merit but does not fully meet PLOS ONE’s publication criteria as it currently stands. Therefore, we invite you to submit a revised version of the manuscript that addresses the points raised during the review process.

The academic editor suggests using 'Type 3 MNV' to replace the term 'retinal angiomatous proliferation or RAP.' Please not the recent publication on consensus nomenclature for reporting neovascular AMD.

Spaide RF, Jaffe GJ, Sarraf D, Freund KB, Sadda SR, Staurenghi G, Waheed NK, Chakravarthy U, Rosenfeld PJ, Holz FG, Souied EH, Cohen SY, Querques G, Ohno-Matsui K, Boyer D, Gaudric A, Blodi B, Baumal CR, Li X, Coscas GJ, Brucker A, Singerman L, Luthert P, Schmitz-Valckenberg S, Schmidt-Erfurth U, Grossniklaus HE, Wilson DJ, Guymer R, Yannuzzi LA, Chew EY, Csaky K, Mones JM, Pauleikhoff D, Tadayoni R, Fujimoto J (2020) Consensus Nomenclature for Reporting Neovascular Age-Related Macular Degeneration Data: Consensus on Neovascular Age-Related Macular Degeneration Nomenclature Study Group. Ophthalmology 127 (5):616-636. doi:10.1016/j.ophtha.2019.11.004

We look forward to receiving your revised manuscript.

Kind regards,

Yuhua Zhang

Academic Editor

PLOS ONE

Journal Requirements:

2. If you are reporting a retrospective study of medical records or archived samples, please ensure that you have discussed whether all data were fully anonymized before you accessed them. If patients provided informed written consent to have data from their medical records used in research, please include this information.

Reviewers' comments:

Reviewer's Responses to Questions

**Comments to the Author**

1. Is the manuscript technically sound, and do the data support the conclusions?

Reviewer #1: Partly

Reviewer #2: No

2. Has the statistical analysis been performed appropriately and rigorously? 

Reviewer #1: Yes

Reviewer #2: Yes

3. Have the authors made all data underlying the findings in their manuscript fully available?

Reviewer #1: No

Reviewer #2: No

4. Is the manuscript presented in an intelligible fashion and written in standard English?

Reviewer #1: Yes

Reviewer #2: Yes

5. Review Comments to the Author

Reviewer #1: I think that the findings could be of interest to the readers and this is one of the largest series reporting type 3 MNVs.

However, I think that there are some issues that need to be solved:

1° - I have some concerns about the CVI analysis.

- Which type of analysis was performed? Based on one b-scan? Which one? Or a 3D CVI analysis? Please specify.

- Which kind of threshold was applied? An auto-local threshold or not? Was there a compensation mode used in the algorithm?

- I have some concerns about the results. Only a value of 2 for the SD in the results of CVI is a little strange. The % expressed by CVI, is useful very variable between subjects, with higher levels of SD (as reported by the results of several papers). Do you have a technical explanation of that?

2° - The main result of the paper is based on the difference in CVI. However, in %, the mean difference between the two groups was only 1% (60.7% vs 61.7%). This is a very little difference and I am not sure that it is clinically relevant and repeatable. How was calculated the sample size? Which margin of error? Which power? Which alpha? Because it is very important to conduct an analysis based on the sample size. In fact, if the sample is too large for the expected results, some difference could be statistically significant only because the sample is too large. This is a very important point because the sample here is large, the difference is small (60.7% vs 61.7%), and the p-value was at the limit of significance (0.047).

3° - I think that is better to refer to type 3 MNV as “Type 3 MNV” (Freund KB, Ho IV, Barbazetto IA, Koizumi H, Laud K, Ferrara D, Matsumoto Y, Sorenson JA, Yannuzzi L. Type 3 neovascularization: the expanded spectrum of retinal angiomatous proliferation. Retina. 2008 Feb;28(2):201-11.) instead of RAP. Indeed, the term “RAP” was coined based on a retinal origin of type 3 MNV. On the other hand, Type 3 MNV is independent of the origin of the lesion (retinal or choroidal origin).

4°- Several in vivo studies, suggested the retinal origin of the type 3 MNV (i.e. Sacconi R, Sarraf D, Garrity S, Freund KB, Yannuzzi LA, Gal-Or O, Souied E, Sieiro A, Corbelli E, Carnevali A, Querques L, Bandello F, Querques G. Nascent Type 3 Neovascularization in Age-Related Macular Degeneration. Ophthalmol Retina. 2018 Nov;2(11):1097-1106.). I think that the authors should cite it.

Reviewer #2: To Authors,

The concept of this paper is slightly interesting.

The study may be the first study for investigating the influence of the CVI in the development of fellow eye neovascularization in patients with unilateral RAP. However, the significant difference to be just “p=0.047”, which should be too low for making people to have common sense.

At line 193-199, the authors mentioned,

“In the current study, we revealed that a lower CVI was associated with the development of neovascularization in the unaffected fellow eye of RAP patients. Reduced choroidal vascularity could lead to choroidal ischemic change, and this condition might affect the formation of choroidal neovascularization. Moreover, this result may suggest that lower CVI values in the unaffected fellow eyes of patients with RAP indicate progressed choroidal ischemic change and possible subclinical disease. Based on the results of the current study, we recommend early evaluation of the CVI in the unaffected fellow eyes of patients with RAP and establishing a proper follow-up plan.”

However, this study showed the significant difference to be just “p=0.047”, which should be too low for making people to have common sense. The authors should rewrite the paragraph.

In the Discussion section, the authors did make no mention of the study of PLoS ONE 15(12): e0243458. https://doi.org/10.1371/journal.pone.0243458, which is the latest publication on the topic about onset of RAP lesions. This paper must be presented and discussed in the Discussion section of the paper by the authors

6. PLOS authors have the option to publish the peer review history of their article (what does this mean?). If published, this will include your full peer review and any attached files.

Reviewer #1: No

Reviewer #2: No

---

## [Author Response · Author response to Decision Letter 0]

11 May 2021

Reviewer #1: I think that the findings could be of interest to the readers and this is one of the largest series reporting type 3 MNVs.

However, I think that there are some issues that need to be solved:

1° - I have some concerns about the CVI analysis.

- Which type of analysis was performed? Based on one b-scan? Which one? Or a 3D CVI analysis? Please specify. 

Response) An OCT B-scan image based and Agrawal’s method was adapted. We add this in the methods section, page 4, line 99.

- Which kind of threshold was applied? An auto-local threshold or not? Was there a compensation mode used in the algorithm?

Response) Niblack auto-local threshold was applied, and other compensation modes were not used. We add this in the methods section, page 5, line 101-102

- I have some concerns about the results. Only a value of 2 for the SD in the results of CVI is a little strange. The % expressed by CVI, is useful very variable between subjects, with higher levels of SD (as reported by the results of several papers). Do you have a technical explanation of that?

Response) Thank you for pointing this out. In this study, we measured CVI according to the method of Agrawal et al, as mentioned in the methods section, and there were no technical differences. And referring to CVI in existing literatures (Rupesh Agrawal, Jianbin Ding, Parveen Sen, Andres Rousselot, Amy Chan, Lisa Nivison-Smith, Xin Wei, Sarakshi Mahajan, Ramasamy Kim, Chitaranjan Mishra, Manisha Agarwal, Min Hee Suh, Saurabh Luthra, Marion R. Munk, Carol Y. Cheung, Vishali Gupta, Exploring choroidal angioarchitecture in health and disease using choroidal vascularity index, Progress in Retinal and Eye Research, Volume 77, 2020,100829), it has been reported that the SD value of CVI varies from less than 1 to more than 10 in various diseases. 

2° - The main result of the paper is based on the difference in CVI. However, in %, the mean difference between the two groups was only 1% (60.7% vs 61.7%). This is a very little difference and I am not sure that it is clinically relevant and repeatable. How was calculated the sample size? Which margin of error? Which power? Which alpha? Because it is very important to conduct an analysis based on the sample size. In fact, if the sample is too large for the expected results, some difference could be statistically significant only because the sample is too large. This is a very important point because the sample here is large, the difference is small (60.7% vs 61.7%), and the p-value was at the limit of significance (0.047). 

Response) Thank you for pointing this out. Since this study was conducted in a retrospectively medical chart review method, the sample size could not be calculated at the research design stage. In fact, there is a lot of debate about whether it makes sense for post-hoc power analysis, and it is said that If a sample is selected, outcomes are no longer random and power analysis becomes meaningless for this particular study sample. (Zhang Y, Hedo R, Rivera A, et. al Post hoc power analysis: is it an informative and meaningful analysis? General Psychiatry 2019;32:e100069. doi: 10.1136/gpsych-2019-100069). Therefore, we have not added any information about post-hoc power analysis to the manuscript. However, we added to the discussion section that one of the limitations of this study was that the mean difference between the CVI results is small and the p-value was close to the upper limit of the statistical significance. (Revised manuscript with tract changes, Discussion, page 10, line 231-235) 

3° - I think that is better to refer to type 3 MNV as “Type 3 MNV” (Freund KB, Ho IV, Barbazetto IA, Koizumi H, Laud K, Ferrara D, Matsumoto Y, Sorenson JA, Yannuzzi L. Type 3 neovascularization: the expanded spectrum of retinal angiomatous proliferation. Retina. 2008 Feb;28(2):201-11.) instead of RAP. Indeed, the term “RAP” was coined based on a retinal origin of type 3 MNV. On the other hand, Type 3 MNV is independent of the origin of the lesion (retinal or choroidal origin).

Response) We very much appreciate this comment and totally agree with you. We have revised entire manuscript and added related information to the introduction section, page 3, line 49-52 in revised manuscript with tract changes.

4°- Several in vivo studies, suggested the retinal origin of the type 3 MNV (i.e. Sacconi R, Sarraf D, Garrity S, Freund KB, Yannuzzi LA, Gal-Or O, Souied E, Sieiro A, Corbelli E, Carnevali A, Querques L, Bandello F, Querques G. Nascent Type 3 Neovascularization in Age-Related Macular Degeneration. Ophthalmol Retina. 2018 Nov;2(11):1097-1106.). 

I think that the authors should cite it

Response) Thank you for the comments. We add this in the introduction section, page 3, line 69-71 in revised manuscript with tract changes.

Reviewer #2: To Authors,

The concept of this paper is slightly interesting.

The study may be the first study for investigating the influence of the CVI in the development of fellow eye neovascularization in patients with unilateral RAP. However, the significant difference to be just “p=0.047”, which should be too low for making people to have common sense.

Response) We very much appreciate this comment and totally agree with you. We added to the discussion section that one of the limitations of this study was that the p-value was close to the upper limit of the statistical significance. (Revised manuscript with tract changes, Discussion, page 10, line 231-235)

At line 193-199, the authors mentioned,

“In the current study, we revealed that a lower CVI was associated with the development of neovascularization in the unaffected fellow eye of RAP patients. Reduced choroidal vascularity could lead to choroidal ischemic change, and this condition might affect the formation of choroidal neovascularization. Moreover, this result may suggest that lower CVI values in the unaffected fellow eyes of patients with RAP indicate progressed choroidal ischemic change and possible subclinical disease. Based on the results of the current study, we recommend early evaluation of the CVI in the unaffected fellow eyes of patients with RAP and establishing a proper follow-up plan.”

However, this study showed the significant difference to be just “p=0.047”, which should be too low for making people to have common sense. The authors should rewrite the paragraph.

Response) Thank you for pointing this out. As per your comments, we have revised the paragraph. (Revised manuscript with tract changes, Discussion, page 9, line 204-212)

---

## [Decision Letter · Decision Letter 1]

14 Jun 2021

PONE-D-21-05303R1

Unaffected fellow eye neovascularization in patients with type 3 neovascularization: incidence and risk factors

PLOS ONE

Dear Dr. Park,

Thank you for submitting your manuscript to PLOS ONE. After careful consideration, we feel that it has merit but does not fully meet PLOS ONE’s publication criteria as it currently stands. Therefore, we invite you to submit a revised version of the manuscript that addresses the points raised during the review process.

I hope you can address the concerns raised by Reviewer 2, though you do not have to agree with the comments. 

We look forward to receiving your revised manuscript.

Kind regards,

Yuhua Zhang

Academic Editor

PLOS ONE

Additional Editor Comments (if provided):

I hope you can address the comments from reviewer 2.

Reviewers' comments:

Reviewer's Responses to Questions

**Comments to the Author**

1. If the authors have adequately addressed your comments raised in a previous round of review and you feel that this manuscript is now acceptable for publication, you may indicate that here to bypass the “Comments to the Author” section, enter your conflict of interest statement in the “Confidential to Editor” section, and submit your "Accept" recommendation.

Reviewer #1: All comments have been addressed

Reviewer #2: (No Response)

2. Is the manuscript technically sound, and do the data support the conclusions?

Reviewer #1: Yes

Reviewer #2: Partly

3. Has the statistical analysis been performed appropriately and rigorously? 

Reviewer #1: Yes

Reviewer #2: Yes

4. Have the authors made all data underlying the findings in their manuscript fully available?

Reviewer #1: Yes

Reviewer #2: No

5. Is the manuscript presented in an intelligible fashion and written in standard English?

Reviewer #1: Yes

Reviewer #2: Yes

6. Review Comments to the Author

Reviewer #1: The authors should be congratulated because they address all the reviewers' comments. I have no more issues.

Reviewer #2: I could not find your comments about the following question.

In the Discussion section, the authors did make no mention of the study of PLoS ONE 15(12): e0243458. https://doi.org/10.1371/journal.pone.0243458, which is the latest publication on the topic about onset of RAP lesions (type 3 MNV). This paper must be presented and discussed in the Discussion section of the paper by the authors.

7. PLOS authors have the option to publish the peer review history of their article (what does this mean?). If published, this will include your full peer review and any attached files.

Reviewer #1: No

Reviewer #2: No

---

## [Author Response · Author response to Decision Letter 1]

16 Jun 2021

Reviewer #1: The authors should be congratulated because they address all the reviewers' comments. I have no more issues.

Response) We very much appreciate your comment.

Reviewer #2: I could not find your comments about the following question.

In the Discussion section, the authors did make no mention of the study of PLoS ONE 15(12): e0243458. https://doi.org/10.1371/journal.pone.0243458, which is the latest publication on the topic about onset of RAP lesions (type 3 MNV). This paper must be presented and discussed in the Discussion section of the paper by the authors.

Response) Thank you for pointing this out. The study mentioned above suggested that melanin deficiency in RPE cells is a result of intensive macular stress and that neovascularization may occur in association with several cytokines secreted during this period. Here, intensive macular stress can be induced from ischemic choroidal condition, so our study is thought to show a trend that is not different from the results of the above study. We add this in the discussion section, page 9, line 191-193 and 208-212 in revised manuscript with tract changes.

---

## [Editor Report · Decision Letter 2]

22 Jun 2021

Unaffected fellow eye neovascularization in patients with type 3 neovascularization: incidence and risk factors

PONE-D-21-05303R2

Dear Dr. Park,

We’re pleased to inform you that your manuscript has been judged scientifically suitable for publication and will be formally accepted for publication once it meets all outstanding technical requirements.

Kind regards,

Yuhua Zhang

Academic Editor

PLOS ONE
---

## [Editor Report · Acceptance letter]

7 Jul 2021

PONE-D-21-05303R2 

Unaffected fellow eye neovascularization in patients with type 3 neovascularization: incidence and risk factors 

Dear Dr. Park:

I'm pleased to inform you that your manuscript has been deemed suitable for publication in PLOS ONE. Congratulations! Your manuscript is now with our production department. 

Kind regards, 

on behalf of

Dr. Yuhua Zhang 

Academic Editor

PLOS ONE